# Distinct neural mechanisms underlie the success, precision, and vividness of episodic memory

**Franziska R Richter[†], Rose A Cooper[†], Paul M Bays, Jon S Simons***

Department of Psychology, University of Cambridge, Cambridge, United Kingdom

**Abstract** A network of brain regions have been linked with episodic memory retrieval, but limited progress has been made in identifying the contributions of distinct parts of the network. Here, we utilized continuous measures of retrieval to dissociate three components of episodic memory: retrieval success, precision, and vividness. In the fMRI scanner, participants encoded objects that varied continuously on three features: color, orientation, and location. Participants' memory was tested by having them recreate the appearance of the object features using a continuous dial, and continuous vividness judgments were recorded. Retrieval success, precision, and vividness were dissociable both behaviorally and neurally: successful versus unsuccessful retrieval was associated with hippocampal activity, retrieval precision scaled with activity in the angular gyrus, and vividness judgments tracked activity in the precuneus. The ability to dissociate these components of episodic memory reveals the benefit afforded by measuring memory on a continuous scale, allowing functional parcellation of the retrieval network.

**\*For correspondence:** jss30@cam.
ac.uk

[†]These authors contributed
equally to this work

**Competing interests:** The
authors declare that no
competing interests exist.

**Reviewing editor:** Lila Davachi,
New York University, United
States

## Introduction

Remembering previous events is one of the hallmarks of human cognition, with memory retrieval contributing to, and depending on, many other cognitive abilities. Memory retrieval involves a complex set of processes that activate a large network of brain regions (*Rugg and Vilberg, 2013*; *Spaniol et al., 2009*; *Kim, 2010*). Recently, it has been suggested that the memory retrieval network can be divided into two sub-systems (*Ranganath and Ritchey, 2012*), both anchored on the hippocampus: an anterior temporal system, comprising the amygdala, temporopolar cortex, orbitofrontal cortex and perirhinal cortex, and a posterior medial system, including regions such as the posterior cingulate, precuneus, angular gyrus (AnG), medial prefrontal cortex, and parahippocampal cortex. According to this framework, the anterior temporal system is more related to semantic memory, familiarity judgments, and event salience, whereas the posterior medial system contributes to episodic memory and recollection. Despite such attempts to characterize sub-systems within the retrieval network, and suggestions about the roles of some of its constituent regions, functional specializations within the memory retrieval network remain poorly understood, in part due to common co-activation during most retrieval tasks.

It is likely that memory retrieval tasks typically activate such a wide network of regions because the components of successful retrieval can reflect the contribution of a number of different properties of memory representations, such as their subjective vividness, objective accuracy, and the objective level of detail, specificity, or precision with which they are retrieved. To better understand how memory judgments are derived, it is important to consider that the outcome of a retrieval attempt is typically not 'all-or-none', but that the amount and quality of information we retrieve can vary considerably (*Parks and Yonelinas, 2007*). In order to distinguish memory retrieval components and the role of regions within the retrieval network, it is necessary to use sensitive measures that can detect

**eLife digest** Remembering is something we do countless times each day. The detail and vividness with which we can remember is part of what makes memories so precious. Given the significance and complexity of memories, it is perhaps unsurprising that several parts of the brain are needed for us to experience them. Indeed, the brain regions involved in memory all work so closely together that it is a challenge to identify what role each region plays.

Richter, Cooper et al. aimed to design a memory task that could separate key characteristics of remembering, which would allow them to study links between each aspect and the different brain regions involved in memory. The resulting test involved showing people images of different objects whilst they were in an MRI medical imaging scanner. The people taking the test were asked to remember several objects that could vary in color, position and orientation. Participants were asked to rate how vividly they remembered the objects and then tried to precisely recreate their color, orientation and position.

The test allowed Richter, Cooper et al. to link specific parts of the brain to certain aspects of remembering. The hippocampus, an area known to be important in memory processing, indicated whether or not information had been remembered. More vivid memories were linked to greater activity in a region called the precuneus, which plays a role in imagination. Lastly, activity in a third region – the angular gyrus – indicated the precision of each memory.

Being able to study different aspects of memory using tests like this that collect detailed measurements could be important in identifying memory problems, for example, in people with brain diseases or head injuries, or after a stroke. Specifically, the methods developed by Richter, Cooper et al. could provide sensitive tools for detecting memory difficulties at an early stage. This may help more people to get treated sooner, potentially minimizing lasting complications.

such variations in retrieval performance. Consistent with this approach, some research has studied the specificity with which memories can be retrieved as an alternative to simple 'old/new' decisions or other categorical judgments such as 'remember/know'. For example, source memory tasks, requiring participants to remember the context in which an item was studied, have revealed that participants sometimes remember 'partial source' information (such as the gender of a speaker), but not necessarily all specific details (*Simons et al., 2004*). The idea that memory representations can vary in quality has prompted researchers to measure recollection objectively using more graded measures, exploring how many details can be remembered (*Horner and Burgess, 2014*; *Qin et al., 2011*; *Vilberg and Rugg, 2009*). Although such measures are important advances in capturing differences in retrieval outcome, they remain essentially categorical. Memory is not only defined by the number of details that are remembered, but also by the fidelity with which those details can be recalled (*Brady et al., 2013*).

Recent developments in the study of short-term memory have demonstrated that memory representations vary along a continuous scale of memory precision (*Bays and Husain, 2008*; *Zhang and Luck, 2008*; *Fougnie et al., 2012*; *van den Berg et al., 2012*; *Bays, 2014*), highlighting the flexibility of memory representations in terms of both quantity and quality (*Ma et al., 2014*). These influential methods from the working memory literature have started to be applied to behavioral studies of long-term memory (*Brady et al., 2013*). Importantly, it is possible to separate behaviorally the probability of recollection success from the precision with which information is retrieved (*Harlow and Donaldson, 2013*; *Harlow and Yonelinas, 2016*), which can be differentially influenced by attention and retrieval practice (*Fan and Turk-Browne, 2013*). Furthermore, a recent EEG study supports the idea that variations in long-term memory precision may be associated with distinct neural signatures, finding the well-established left parietal old/new effect to be graded according to recollection precision (*Murray et al., 2015*). The observation that retrieval success and retrieval precision can be manipulated independently suggests that using continuous measures to differentiate such components will lead to a more detailed understanding of episodic memory retrieval at both behavioral and neural levels.

Retrieval success and retrieval precision both constitute objective measures of memory performance. However, increasing research has additionally considered subjective or metacognitive measures of memory performance, such as confidence (*Yonelinas, 1994*) or the vividness with which retrieved memories are experienced (*Kuhl and Chun, 2014*; *St-Laurent et al., 2015*). Research has found that subjective and objective measures of memory performance can diverge (*Chua et al., 2012*; *Harlow and Yonelinas, 2016*). Most notably, patients with parietal cortex lesions exhibit impairment in subjective memory reports even though their objective performance appears to be intact (*Simons et al., 2010*), and lateral parietal transcranial magnetic stimulation has been found to selectively reduce memory confidence but not retrieval success (*Yazar et al., 2014*). One possibility is that the subjective and objective memory components that may drive mnemonic decisions are separable, with subjective aspects of retrieval, such as vividness, and objective aspects of retrieval, such as success and precision, relying on largely independent processes. Alternatively, it is possible that performance on subjective measures of memory retrieval used to date can be fully explained by subtle variations in objective memory, such as the precision with which a memory is recalled, which previous objective measures may not have been sensitive enough to detect.

To summarize, memory retrieval involves a wide network of brain regions and may reflect the contribution of both objective and subjective memory components, with objective memory success being distinguishable from objective memory precision and both being distinguishable from subjective memory vividness. To understand the roles of brain regions within the retrieval network and to dissociate memory retrieval components, more sensitive, continuous measures of memory performance are needed. In the current study, we combined continuous long-term memory measures in an fMRI paradigm with model-based approaches derived from the working memory literature. For each retrieval trial we obtained a binary measure of retrieval success (successful versus unsuccessful), a continuous measure of the precision with which color, orientation, and location features of visual objects were remembered (based on the error between a target value and the participant's response), as well as a subjective vividness rating, on a continuous scale. These measures allowed us to characterize behavioral and neural mechanisms that distinguish between recollection success, precision, and the vividness with which information is retrieved.

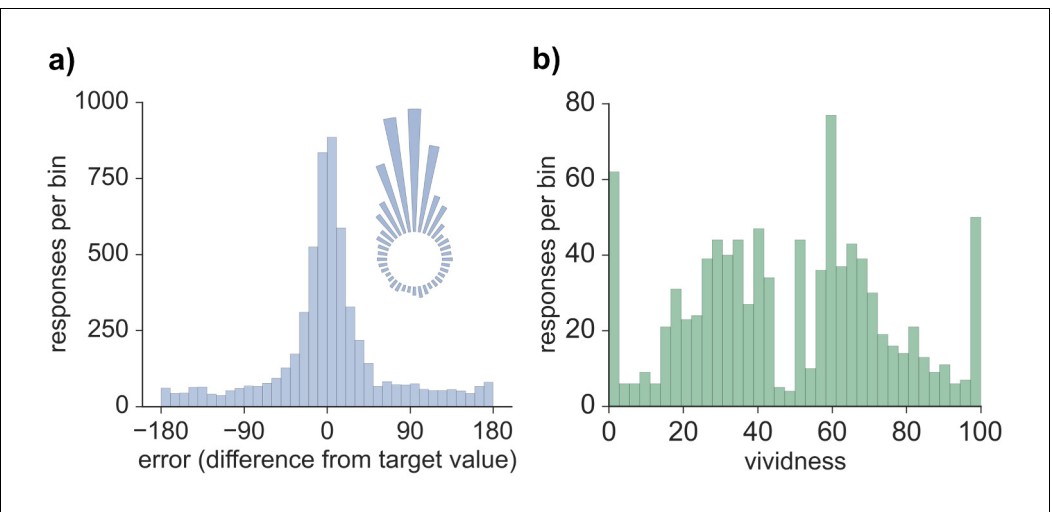

**Figure 1.** Behavioral responses for all retrieval trials. (a) Distribution of errors (difference between reported feature value and target feature value), across all 5724 trials across all participants, also visualized in circular space by wrapping the distribution around a circle. (b) Distribution of vividness responses across all 954 vividness ratings for all participants.
The following source data is available for figure 1:

**Source data 1.** Data associated with the error distribution and vividness rating analyses.

## Results

### Behavioral results

Model-based analyses were first conducted on the error values obtained for each memory retrieval trial. The distribution of errors for all 5724 trials across all participants and features (see Materials and methods) can be seen in *Figure 1*. Three models were fitted to the error data (*Bays et al., 2009*), both across all trials as well as separately for each participant, and model fit was assessed via Akaike information criterion (AIC) and Bayesian information criterion (BIC). Modeling the errors with a von Mises (circular normal) distribution alone (model 1) fit the data less well than a mixture model (model 2) that also included a uniform component (AIC difference: 1882.2; BIC difference: 1875.6; also true across subjects, AIC: $t(19) = 9.01$, $p < 0.001$; BIC: $t(19) = 8.63$, $p < 0.001$) and a third model in which non-target binding errors were additionally accounted for (AIC difference: 1880.2; BIC difference: 1866.9; also true across subjects, AIC: $t(19) = 8.96$, $p < 0.001$; BIC: $t(19) = 8.20$, $p < 0.001$). Model 2 provided a better fit than model 3 (AIC difference: 2.0; BIC difference: 8.7; also true across subjects, AIC: $t(19) = 4.61$, $p < 0.001$; BIC: $t(19) = 16.48$, $p < 0.001$), suggesting that non-target errors did not account for a significant amount of variance in performance on this long-term memory task. Therefore, the distribution of errors was best characterized by a model with one parameter capturing the proportion of successfully retrieved features (responses within the von Mises distribution relative to the uniform distribution) and another parameter capturing the precision (concentration) of successfully retrieved memories. As seen in *Figure 1*, errors clustered around the target value, indicating that participants often successfully retrieved information about the probed feature. Moreover, a certain percentage of trials resulted in errors evenly distributed between −180 and 180 degrees, consistent with responses due to forgetting. Using model 2 (von Mises + uniform), the estimated mean proportion of trials successfully retrieved was 0.62 (*SD* = 0.16), and estimated mean precision (assessed by the concentration parameter of the von Mises distribution) was 10.05 (*SD* = 3.41). The distribution of vividness responses across all subjects and trials can also be seen in *Figure 1*. We calculated the mean vividness across all 48 displays for each subject (*mean* = 48.23, *SD* = 15.83), which indicated that subjects' memories were, on average, moderately vivid, and responses were widely spread between 0–100.

We hypothesized that our three indices of memory retrieval (retrieval success, precision and vividness) measure different components of retrieval performance. If so, we would expect them to correlate, but only to a moderate degree. To compare the covariance of retrieval success, precision, and

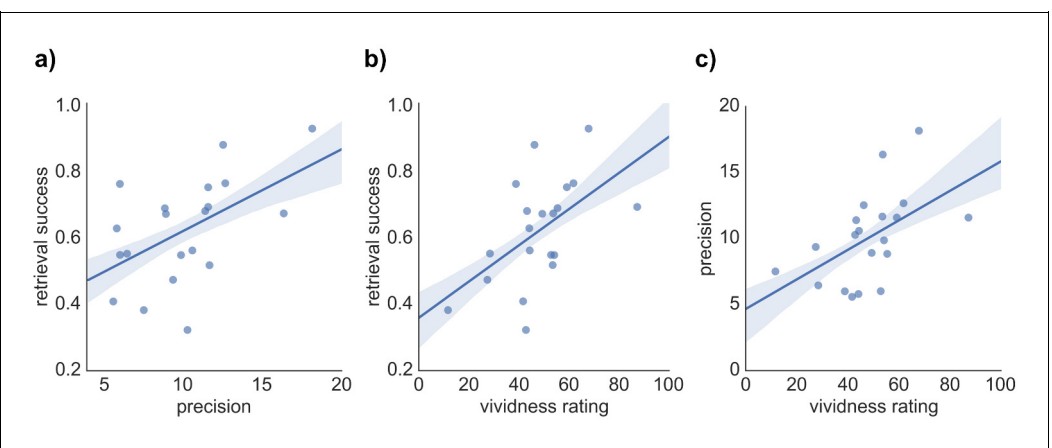

**Figure 2.** Relationship between the three measures of retrieval performance. Correlation between (**a**) retrieval success and precision, (**b**) retrieval success and vividness ratings, and (**c**) precision and vividness ratings. Shaded areas indicate the standard error of the correlation.
The following source data is available for figure 2:

**Source data 1.** Data associated with the pairwise correlation analyses.

vividness, across-subject pairwise correlation analyses and separate regression analyses predicting each retrieval measure were conducted. As expected, each pairwise correlation revealed a moderate positive correlation of similar magnitude, including a relationship between retrieval success and precision ($r = 0.527$, $p = 0.017$), retrieval success and vividness ($r = 0.543$, $p = 0.013$), and precision and vividness ($r = 0.519$, $p = 0.019$) (see *Figure 2*). However, regression analyses highlighted that a substantial proportion of variance in each retrieval measure remained unexplained when predicted by the other two measures: For retrieval success, 62.3% of the variance was unexplained after accounting for vividness and precision ($R^2 = 0.377$, $F(2,17) = 5.14$, $p = 0.018$, with no significant individual effect of vividness, $\beta = 0.369$, $t(17) = 1.65$, $p = 0.118$, or precision, $\beta = 0.336$, $t(17) = 1.50$, $p = 0.152$, when entered in the same model). Similarly, 64.5% of variance in retrieval precision was unexplained by the other two variables ($R^2 = 0.355$, $F(2,17) = 3.67$, $p = 0.024$, with no significant effect of vividness, $\beta = 0.330$, $t(17) = 1.42$, $p = 0.173$, or retrieval success, $\beta = 0.348$, $t(17) = 1.50$, $p = 0.152$). Lastly, 63% of variance in retrieval vividness was unexplained by retrieval success and precision ($R^2 = 0.370$, $F(2,17) = 4.98$, $p = 0.020$, with no significant effect of retrieval success, $\beta = 0.373$, $t(17) = 1.64$, $p = 0.118$, or precision, $\beta = 0.322$, $t(17) = 1.42$, $p = 0.173$). The behavioral analyses, therefore, support the relative separability of retrieval success, precision, and vividness at a cognitive level.

## fMRI results

### Trial-specific measures of retrieval success, precision, and vividness

For the purpose of fMRI analyses, we developed trial-specific measures of retrieval success, precision, and vividness. Using a model-derived cut-off for successful retrieval ( +/- 63 degrees from the target value, see Materials and methods: Behavioral analysis), participants achieved an average of 76.04% (SD = 10.74%) successfully retrieved trials. (This percentage correct is higher than that reported for the behavioral analysis because the model adjusts for responses that can be attributed to the uniform distribution.) Trial-specific estimates of precision were obtained by subtracting the absolute difference between the target value and the reported value (ranging from 0–180 degrees) from 180 (the maximum possible error), so that higher values reflected higher precision. Precision was only considered for successfully retrieved trials, as the precision of trials in which retrieval was unsuccessful is likely to reflect guessing, leading errors to be randomly distributed. (Therefore, the actual range of values that was used for the analysis of the precision effects was 117–180.) Lastly, the vividness ratings (0–100) for each display in the retrieval phase were used as a trial-specific measure of subjective vividness for each of the 48 displays.

### ROI activity and conjunction analyses

The three main contrasts of interest reflected the degree to which neural activity related to successful retrieval (successful > unsuccessful), the precision of successful retrieval (increase in BOLD (blood-oxygen-level dependent) signal with increase in precision), and the vividness of retrieval (increase in BOLD signal with increase in vividness). We focused our main fMRI analysis on three a priori regions of interest (ROIs) from the posterior-medial recollection network (*Ranganath and Ritchey, 2012*) that have consistently been activated in the recollection literature: hippocampus, AnG, and precuneus. We focused on regions in the left hemisphere only, as previous results have indicated that such posterior retrieval effects are typically left lateralized (*Spaniol et al., 2009*).

We hypothesized that our three retrieval effects of interest would be associated with distinct patterns of brain activity. Consistent with this prediction, our analysis suggested that all three retrieval effects were neurally dissociable: the contrast between successful and unsuccessful retrieval elicited increased activity in hippocampus (peak: −33, −12, −18, $t(19) = 4.34$, $p = 0.015$, $d = 0.97$), but no significant effects were detected in precuneus (all $t(19) < 3.02$, all $p > 0.362$) or AnG (all $t(19) < 2.38$ all $p > 0.371$). In contrast, trial-by-trial measures of precision scaled with activity in AnG (peak: −54, −54, 33, $t(19) = 6.17$, $p = 0.001$, $d = 1.38$), but no such effects were observed in hippocampus (all $t(19) < 1.80$, all $p > 0.657$) or precuneus (all $t(19) < 2.30$, all $p > 0.760$). Lastly, trial-by-trial vividness ratings correlated with brain activity in precuneus (peak: −3, −57, 48, $t(19) = 4.60$, $p = 0.032$, $d = 1.03$), but neither effects in the hippocampus (all $t(19) < 3.54$, all $p > 0.071$) nor AnG (all $t(19) < 2.77$, all $p > 0.246$) reached significance. *Figure 3(a)* shows the activation in the three ROIs for contrasts that yielded a significant effect.

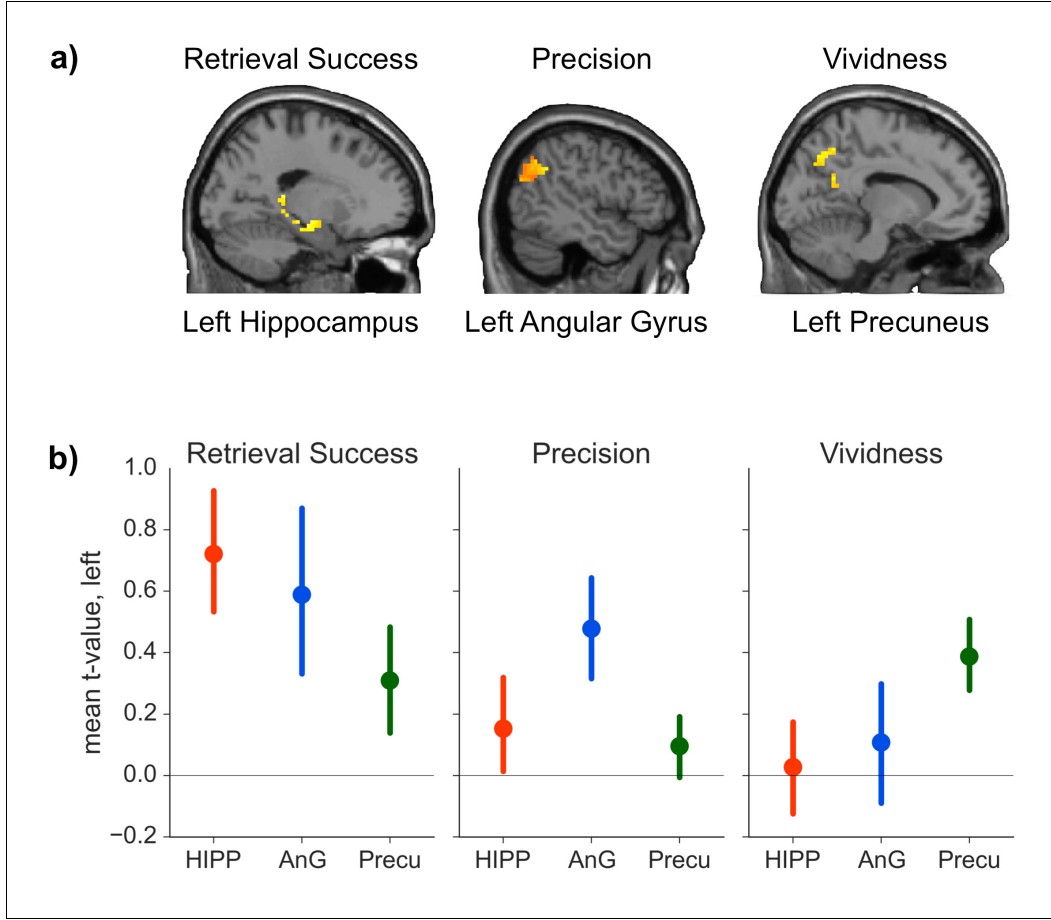

**Figure 3.** ROI analyses. (**a**) Significant effects of the general linear model (GLM) as revealed by the ROI analyses. Each contrast was assessed in each of the *a priori* defined ROIs by determining the significance of peak ROI activity using small-volume correction at p < 0.05. Significant effects of retrieval success were found in the hippocampal ROI, of precision in the AnG ROI, and of vividness in the precuneus ROI. (Contrasts displayed at p < 0.01, uncorrected, for visualization.) (**b**) Mean first level *t*-values observed across voxels in each anatomical ROI for the three measures of retrieval performance. Error bars show standard error of the mean.

The following source data is available for figure 3:

**Source data 1.** Data associated with the ROI analyses.

The results outlined above give some indication that retrieval success, precision and vividness might activate non-overlapping brain areas. To directly test whether the activations observed in the three ROIs for the three contrasts differed in a statistically reliable manner, we conducted a 3 (ROI: hippocampus, AnG, precuneus) × 3 (contrast: retrieval success, precision, vividness) ANOVA on the first level *t*-values for each contrast from each ROI. (First level *t*-values were used because the parameter estimates for the retrieval success contrast reflect the mean difference in BOLD signal, whereas the parameter estimates for the parametric modulators reflect the rate of change of BOLD signal with precision and vividness, meaning that the beta values are not directly comparable). The ANOVA revealed a significant interaction between contrast and ROI ($F_{Huynh-Feldt}(2.973, 56.482) = 4.02$, p = 0.012, $\eta^2 = 0.05$, see *Figure 3b*), indicating that the pattern of regional activity differed significantly between the contrasts. To explore this interaction further, three 2 × 2 ANOVAs were conducted between each pair of contrasts and the region that had the strongest associated activity for each contrast. All three ANOVAs revealed significant interactions between ROI and contrast (all $F > 4.63$, all $p < 0.05$, all $\eta^2 > 0.03$), further specifying the distinct roles of the ROIs in the memory retrieval components of interest. Conducting conjunction analyses within the ROIs for each of the

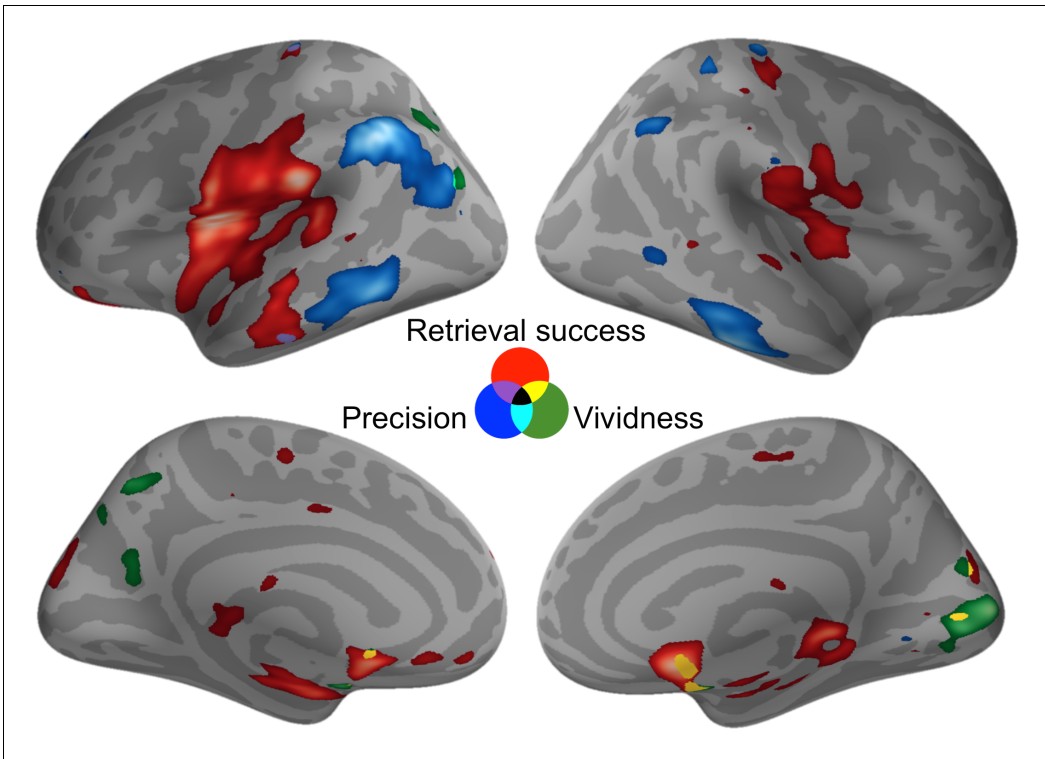

**Figure 4.** Activity for each contrast of interest. Activity was assessed throughout the brain to reveal any areas of overlap (at a liberal statistical threshold of p < 0.001, uncorrected). Activations are displayed on the cortical surface using Pysurfer software (https://pysurfer.github.io), visualized at an even more liberal threshold of p < 0.01 (uncorrected). Red = Retrieval Success; Blue = Precision; Green = Vividness. Greater activity is indicated by increasing brightness of the color. Purple = overlap between Retrieval Success and Precision; Cyan = overlap between Precision and Vividness; Yellow = Overlap between Retrieval Success and Vividness.

three pairs of contrasts revealed no voxels in which there was any significant overlap even at a liberal threshold using a conjoint probability of p < 0.001 (uncorrected) (each individual contrast set to a threshold of p < 0.01), consistent with the findings of significant activity in only one ROI per contrast.

The analyses described thus far provide evidence for a dissociation in the neural basis of retrieval success, precision, and vividness in the ROIs selected a priori, but it is possible that activity relating to these memory components might overlap elsewhere in the brain. To test for similarities in neural activity, three whole-brain pairwise conjunction analyses were conducted between each pair of retrieval measures using a liberal conjoint probability threshold of p < 0.001 (uncorrected) with a minimum cluster size of at least 20 contiguous voxels (see *Figure 4*). Conjunction analyses between retrieval success and precision, and between precision and vividness, revealed no significant areas of overlap. The conjunction between retrieval success and vividness revealed one significant cluster of overlapping activity in the right caudate (peak: 12, 6, −9, cluster size = 36, $t(19)$ = 5.44, p = 0.033). This very limited amount of overlap at the whole-brain level is consistent with our finding of dissociable effects within the a priori ROIs. Hence, these three memory components appear to be supported by distinct regions within the memory retrieval network.

## Discussion

Many cognitive components contribute to successful recollection, supported by a complex network of brain regions; however, cognitive and neural dissociations within the episodic retrieval network remain underspecified. Here, we sought to distinguish the neurocognitive mechanisms underlying retrieval success, retrieval precision, and retrieval vividness using model-based analyses of

continuous episodic retrieval measures. Retrieval success (successful vs. unsuccessful) was associated with BOLD signal increases in hippocampus, whereas retrieval precision scaled with brain activity in AnG. Both these objective measures of retrieval performance were dissociable from a subjective continuous measure of retrieval vividness, which was tracked by activity in the precuneus. The apparently separable contribution of these three nodes of the episodic retrieval network was confirmed by significant statistical interactions between region and component, and by the lack of overlap in activity revealed by conjunction analyses. Together, the observed behavioral and neural dissociations, revealed by the use of continuous memory measures, shed new light on the distinct contributions made by components of the episodic retrieval network.

Using continuous measures of retrieval performance in addition to the traditional categorical distinction of retrieval success, we were able to dissociate the functions of regions that are often co-activated during memory retrieval. The hippocampus has received most attention in the literature as a 'memory region', because of the profound memory deficits associated with hippocampal damage (*Scoville and Milner, 1957*), and the frequent reports of hippocampal activity during neuroimaging studies of memory retrieval (*Schacter and Wagner, 1999*). One proposal in the cognitive literature has been that hippocampal activity might relate to a threshold retrieval signal (*Yonelinas, 2002*), such that items only elicit a hippocampal recollection response once a specific qualitative information threshold is reached. Consistent with this view, computational models of hippocampal function have produced thresholded output, such that some items produce a memory signal strong enough to elicit an 'old' response, whereas items that remain below the required threshold are indistinguishable to the model from new items (*Norman, 2010*). fMRI results have supported the view that hippocampal activity during retrieval is characterized by distinct states, such that it responds to recollection responses, for example in source memory tasks (*Ranganath et al., 2004*), but not graded familiarity judgments (*Montaldi et al., 2006*). Our observation of a categorical retrieval success effect in the hippocampus is thus consistent with these prior results, but our findings provide novel constraints on understanding of the specificity of the hippocampal role in episodic memory by demonstrating a disproportionately greater role in retrieval success than graded precision or subjective vividness.

In contrast to successful versus unsuccessful retrieval, the precision with which recollected information is retrieved was tracked by activity in AnG, consistent with recent EEG evidence that the late lateral parietal ERP effect is graded according to memory precision (*Murray et al., 2015*). This dissociation between hippocampus and AnG is consistent with the notion of a strongly interconnected cortico-hippocampal network in which hippocampus initiates retrieval, which is then further supported by cortical regions (*McClelland et al., 1995*), with increasing connectivity between different nodes in the network being related to superior mnemonic performance (*Wang et al., 2014*). The specific dissociation between hippocampus and AnG observed here is also consistent with recent reports that hippocampal signals are transient during memory retrieval, as might be expected of a threshold signal, whereas AnG responses are more sustained over the retrieval period, possibly consistent with a role in the active representation of information (*Vilberg and Rugg, 2012*). The various roles attributed to inferior parietal regions such as AnG during memory retrieval include attentional capture by retrieved information (*Cabeza et al., 2008*), the accumulation of evidence in memory (*Wagner et al., 2005*), or a 'buffer' that stores information for further evaluation during long-term memory retrieval (*Vilberg and Rugg, 2008*). Consistent with the latter account, recent work supports a role for AnG in carrying specific details of memory representations, finding that episodic memories are represented individually and can be decoded by multivariate classifier analyses in this area (*Bonnici et al., 2016*; *Kuhl and Chun, 2014*). These findings indicate that AnG may represent the information on which a mnemonic decision is based, with properties of the representation such as its level of detail or the fidelity of the information retrieved driving decisions about a memory's origin and other associated features. This representation of detailed memory features might uniquely qualify AnG for demanding mnemonic computations like precision judgments or the integration of complex retrieved features, which hippocampus might instead only support indirectly by initiating and orchestrating other mnemonic processes (*Vilberg and Rugg, 2012*). Additionally, the observation of trial-to-trial variation in AnG activity, correlated with retrieval precision, is consistent with proposals from the working memory literature that memory fidelity may vary from one retrieval instance to the next (*Fougnie et al., 2012*; *van den Berg et al., 2012*), and specifically that memory precision may be determined by the gain of neural activity encoding a stimulus (*Bays, 2014*).

In addition to the dissociation between objective measures of memory retrieval involving the hippocampus and AnG, our subjective measure of memory performance, retrieval vividness, correlated with precuneus activity, consistent with recent findings linking activity in dorsomedial brain areas with how vividly video clips were recalled (*St-Laurent et al., 2015*). In line with a proposed involvement in vividness judgments, previous evidence has found greater connectivity between prefrontal cortex and precuneus during metacognitive decisions concerning retrieved memories (*Baird et al., 2013*), supporting a role for medial parietal regions in assessments of subjective memory quality. Furthermore, a correlation between precuneus structural volume and recollection from a first-person perspective (*Freton et al., 2014*) is consistent with a contribution to reinstating and maintaining a vivid subjective memory representation. The involvement of the precuneus in facilitating subjective memory vividness could explain its common activation during memory retrieval tasks, such as tests of source memory, that often require retrieval of rich and vivid episodic information (*Lundstrom et al., 2005*). It has been proposed that activity in the precuneus reflects the requirement for mental imagery during retrieval (*Fletcher et al., 1995*). The function of the precuneus might thus be complementary to that of the hippocampus, which may be involved in the initial aspects of a proposed 'scene construction' process that occurs when a memory is retrieved, but not in the maintenance of the scene in working memory following construction (*Zeidman et al., 2015*). In accordance with such a distinction, our current data are consistent with the idea that the hippocampus might play an initial role in driving the reconstruction process, but that further translation into an imageable, first person perspective event occurs via precuneus (*Burgess et al., 2001*).

It is worth noting that previous studies have also linked activity in AnG with rated vividness (*Bonnici et al., 2016*; *Kuhl and Chun, 2014*) and confidence (*Qin et al., 2011*), and that hippocampal activity, too, has been reported to be associated with vividness ratings (*Ford and Kensinger, 2016*; *Gilboa et al., 2004*). Moreover, patients with lesions that include the lateral parietal lobe often report reduced memory detail and diminished recollection vividness (*Berryhill et al., 2007*; *Simons et al., 2010*). At first glance, our results might appear to diverge from these prior reports. However, we believe our findings help to distinguish the contributions of precuneus, hippocampus, and AnG to subjective aspects of recollection, by capitalizing on two specific advantages of our study. In previous work, vividness ratings were often recorded *after* the mnemonic decision, meaning that judgments could reflect the participants' perception of how they had performed on that trial; moreover, vividness and precision or retrieval success effects were not distinguished, such that vividness is likely to have been confounded with precision and/or retrieval success. In the current study, making the vividness rating *before* any decision on the objects ensured that it reflected participants' assessment of their subjective ability to bring to mind the appearance of the objects on the display, rather than the influences of perceived difficulty or post-decision processing (*Siedlecka et al., 2016*). Second, we modeled vividness, retrieval success and precision within the same general linear model, meaning that they accounted for independent sources of variance. While our three memory measures were clearly dissociable, they were not completely independent from each other (and in fact were correlated at $r = \sim0.5$ across subjects in the current study). This moderate correlation would imply that trials which were more precisely and/or successfully remembered would be, on average, associated with higher vividness ratings, possibly explaining previous reports of a link between vividness ratings and brain activity in AnG (cf. *Bonnici et al., 2016*; *Kuhl and Chun, 2014*), or hippocampus (*Ford and Kensinger, 2016*; *Gilboa et al., 2004*). Regarding reports of diminished vividness following parietal cortex lesions, such lesions are frequently not selective to *lateral* parietal cortex (*Berryhill et al., 2007*), often including medial parietal areas. Moreover, lateral and medial areas are heavily interconnected (*Zhang and Li, 2012*), such that lateral parietal lesions could to some extent affect medial parietal function, by resulting in diminished input into medial parietal regions. This diminished input to medial parietal areas could affect judgments of memory vividness, because the information necessary for the maintenance of a vivid mental scene might be less readily available. Thus, our current data suggest that medial parietal regions play a role in evaluating the perceived vividness of participants' memories, a judgment that may be based on the level of detail or precision of the memory represented in AnG, although further research is still required to tease apart the relative contributions of these regions and elucidate the processes underlying judgments of memory vividness and confidence.

## Conclusion

By capitalizing on the use of continuous measures of objective and subjective retrieval performance, our study provides novel insights into dissociable components of memory retrieval. We demonstrate that regions of the episodic memory network that are normally co-activated during retrieval can be dissociated and linked to specific aspects of retrieval performance. Using the current approach to study and analyze behavioral and fMRI memory data, future studies will be able to characterize more accurately the mechanisms involved in different retrieval tasks, and to assess specific variations in memory performance in target groups, such as patients and older adults.

# Materials and methods

### Participants

Twenty-one subjects (9 female; mean age = 29.81) participated in the current experiment. Subjects were 18–45 years of age, right-handed, fluent English speakers, had normal or corrected-to-normal vision, and had no history of psychiatric or neurological disorders. Subjects were recruited via participant databases at the Memory Laboratory, Cambridge University, as well as social media adverts. Informed consent was obtained according to procedures approved by the Cambridge Psychology Research Ethics Committee and subjects were paid a standard honorarium for their time. One male participant was excluded from all analyses due to very low, chance-level behavioral performance (more than 2 *SD* below the group mean for all parts of the memory test), resulting in a final sample of 20 subjects. For one additional subject, the first scan run out of 8 had to be excluded due to a technical error during scanning.

### Materials

Stimuli consisted of 48 background scenes and 144 object pictures. The background scenes consisted of scene photographs (including naturalistic scenes, buildings, and other landmarks), excluding any images showing people or distinct objects. The object stimuli were a subset of photographs of everyday objects used by *Brady et al. (2013)*, obtained from http://timbrady.org/stimuli/ColorRotationStimuli.zip. Objects showing rotational symmetry or that were strongly associated with a specific color were excluded. Background scenes and object pictures were randomly combined to create 48 stimulus displays (750 × 750 pixels), each consisting of a unique background scene with 3 unique superimposed objects varying in color, orientation, and location (on an invisible circle). Colors, orientations, and locations of all objects were selected at random from circular parameter spaces with 360 increments (appearing continuous), but with the following constraint: a minimum distance of 62.04 degrees was enforced between the same feature of different objects in the same display. This was the minimum distance required to ensure that the objects would never overlap in their locations and, for consistency, the same minimum distance in feature space was enforced for all features. All object-background assignments were randomized, but the displays were only generated once and were then kept constant across subjects, for whom the order of presentation was randomized.

### Procedures

Upon arrival, participants completed a short training session to familiarize themselves with task requirements, and the continuous response options. Stimuli used in this training session were unique and did not overlap with those used in the main experiment. The fMRI session involved 8 scan blocks, each of which contained an encoding phase and a retrieval phase. The trial structure of the encoding and retrieval phases is displayed in *Figure 5*. A video of an example retrieval trial can be accessed online (*Video 1*).

In each encoding phase, subjects were presented with 6 stimulus displays, each of which was displayed for 12 s. Subjects were instructed to try to learn the appearance of the objects on the display, including their color, orientation, and location on the background, as best as they could. Displays were separated by a fixation cross of varying duration (400 to 2800 ms, approaching a Poisson distribution, with a mean of 961.1 ms). A 10 s delay followed the encoding phase before the retrieval phase started. In each retrieval phase trial, participants were presented first with only the background scene of a display and were asked to rate how vividly they remembered the appearance of

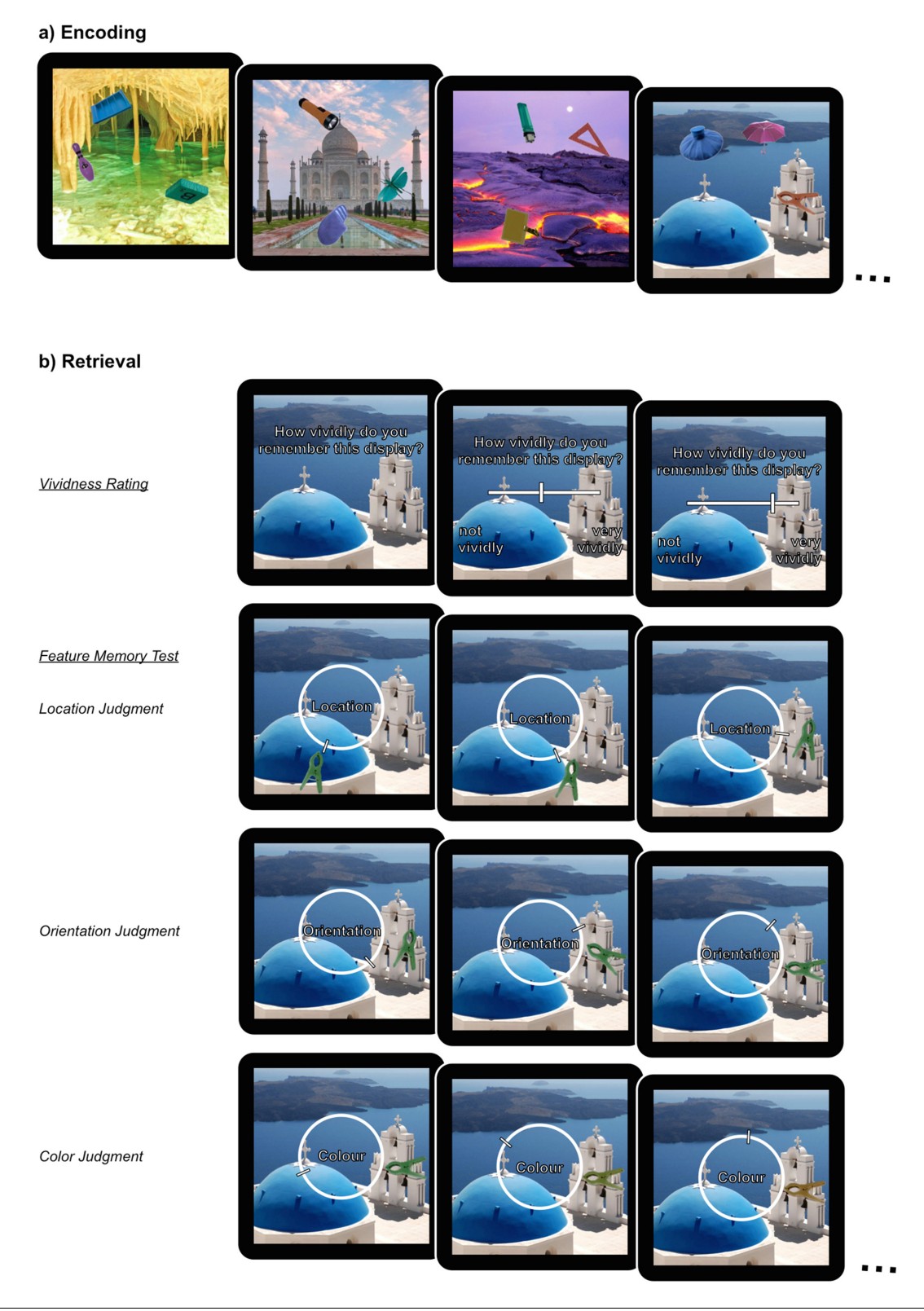

**Figure 5.** Task design. (**a**) Examples of displays learnt during encoding and (**b**) part of the sequence of retrieval questions for a single display illustrating the manipulation of the objects with the continuous dial. During retrieval, the order in which features (color/orientation/location) were tested was counterbalanced. Two out of the three objects associated with an encoded display were tested during retrieval. A video of an example retrieval trial can be accessed online (***Video 1***).

the objects associated with the presented background, based on their memory for the identity and features of all three objects. The background was presented for 2 s with the question "How vividly do you remember this display?" in white font. After 2 s, a slider was added with the labels 'not vividly' and 'very vividly' at the end points. Subjects had 6 s to respond using their index and middle finger to indicate the vividness of their memory on a continuous scale of 0 ('not vividly') to 100 ('very vividly'). They confirmed their responses by pressing a third button with their thumb. In cases where participants did not respond within the first 4 s, the text turned red to indicate to participants that they had 2 s left in which they could respond. The vividness scale was represented by a 500 pixel wide line, on which participants could move a cursor up or down by holding down one or the other of two buttons. Each 'press' of the button moved the cursor by 5 pixels, resulting in the 100-point scale. The vividness rating was requested from participants before they recreated any of the object features so that their subjective vividness judgment would not be influenced by their experience of their performance on that trial. Participants were then tested on their memory for two of the three objects that were associated with a given display, with the same two objects tested for all participants. Each tested object was initially presented in a random color, location, and orientation. The participants' task was to sequentially recreate the objects' original features (color/orientation/location) using a continuous dial (360 increments) over 6 judgments (3 features for 2 objects). For each judgment, a cue word in the center of the screen instructed the participant which object feature they were being asked to recall ('Color', 'Orientation', or 'Location'). Participants had 6 s to recreate these object features as precisely as they could using their index and middle finger to change the features on a continuous dial. Similar to the vividness rating, the color of the cue word changed from white to red if no response was made within the first 4 s of the 6 s response window. The order in which the features were tested was counterbalanced, such that (1) each of the 6 possible orders of the feature questions was tested equally and (2) the same feature was not tested more than 4 times in a row at the same position (first, second, or third). Once participants recreated one of the object features, the feature value chosen (from 0–360 around the circular space) would be carried over to the subsequent questions for that object. A fixation cross of varying length (400 to 2800 ms, approaching a Poisson distribution, with a mean of 961.1 ms) was displayed after the vividness question as well as after each of the object feature questions.

## Behavioral analysis

To generate estimates of retrieval success and precision, three models were fitted to the distribution of errors (absolute difference between the correct response from 0–360 and the participant's response on each trial) across all the feature questions within each subject (see *Figure 6b–d*): (1) a von Mises distribution; (2) a von Mises + uniform distribution, and (3) a von Mises + uniform distribution + von Mises distributions capturing non-target errors (*Bays et al., 2009*; *Zhang and Luck, 2008*; *Harlow and Donaldson, 2013*; model analysis code available at http://www.paulbays.com/code/JV10/). Model 1 assumes participants' memories continuously vary in precision from 0–180 degrees (with no guessed responses), model 2 assumes that participants either remember a feature with varying precision or guess, resulting in a uniform distribution of responses around the circle, and model 3 assumes that, in addition to model 2, participants also commit binding errors where they remember the color/orientation/location that was present in one of the two other objects in the display, but not the target object. Retrieval success was estimated as the proportion of responses that fell within the von Mises distribution, and precision was estimated as the concentration of the von Mises distribution. Importantly, using this operationalization, retrieval success and precision can vary independently, so that high retrieval success does not necessarily imply a participant will also have high retrieval precision (see *Figure 7* for simulated data that illustrates this dissociation). Of note, when the three models were fitted within each feature separately, assessment of the model fit revealed the same pattern as for all features.

## fMRI methods
### fMRI acquisition
fMRI scanning was performed using a 3T Siemens TIM Trio scanner at the Medical Research Council Cognition and Brain Sciences Unit (MRC-CBSU) in Cambridge, using a 32 channel head coil. Structural images were obtained using a T1-weighted protocol (256 × 256 matrix, 192 1-mm sagittal

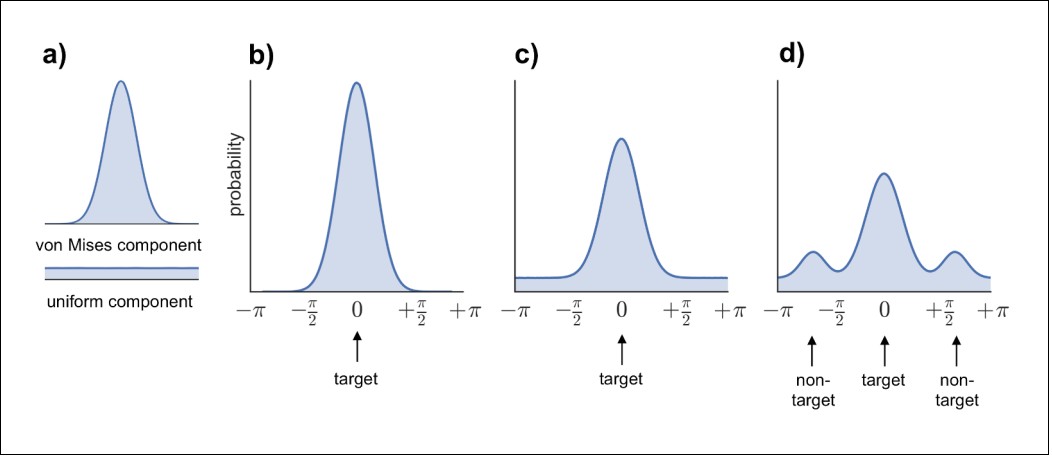

**Figure 6.** Models fitted to the error distribution. (a) Components used to model the error data. (b–d) Illustrations of the three models compared to analyze participants' error distributions, including von Mises alone (b), von Mises + uniform (c), and von Mises + uniform + additional von Mises distributions modeling non-target errors (d).

slices, TR = 2.25 s, TE = 3 ms). Functional images were acquired approximately parallel to the AC–PC transverse plane using a single-shot EPI sequence (TR = 2 s; TE = 30 ms; field of view = 192 × 192 mm, flip angle = 78 degrees). Functional scans were obtained as 32 contiguous oblique-axial slices (3 × 3 × 3-mm voxels) per volume. Each fMRI scan run (block) consisted of one encoding phase followed by one retrieval phase and a total of 205 volumes were collected (6 m 50 s) during each block. ITIs were jittered within each block so that the total duration of each block was the same. Of the 205 volumes, the first 5 were accompanied by a 'Get Ready' screen and were discarded prior to analyses. The next approximately 39 volumes corresponded to the encoding phase. Following the encoding phase, another short break including a 'Get Ready' screen was included for the duration of 5 volumes, and the final approximately 156 volumes corresponded to the retrieval phase.

## fMRI preprocessing

Data preprocessing and analysis was performed using SPM12 (Wellcome Department of Imaging Neuroscience, University College London, London, UK), implemented via an automatic analysis pipeline (version 4; http://www.automaticanalysis.org) as well as custom Matlab scripts. For each

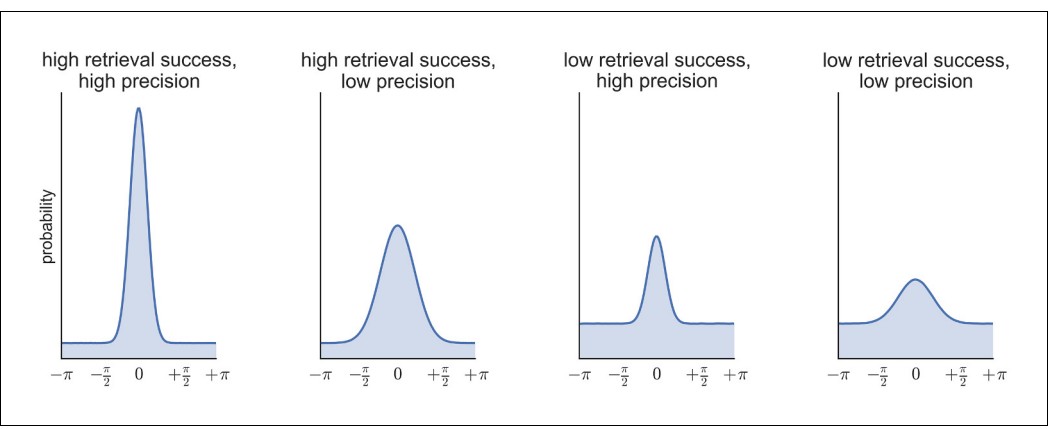

**Figure 7.** Simulated data illustrating the possibility of statistical independence of retrieval success (proportion of responses in the von Mises distribution vs. the uniform distribution) and retrieval precision (the concentration of the von Mises distribution).

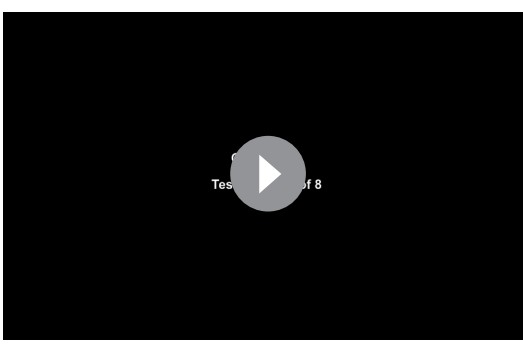

**Video 1.** Example of a retrieval trial. The video demonstrates the use of the continuous response options for the vividness question (00:01–00:09), and the three feature questions for the first (00:09–00:29), as well as the second object (00:30–00:51). Participants were able to adjust their responses continuously by moving around the slider (vividness question) or the circular dial (feature questions). Once they were happy with their response, they locked their response with a button press. For the first five feature questions in the example the participant entered their response within the first 4 s of the question, and therefore the font color of the cue word ('Orientation'/'Colour'/'Location') remains white. For the last feature question, the color of the cue changes to red as no response was given within the first 4 s of the question interval, indicating to the participant that 2 s remain to respond.

participant the structural image was coregistered to the SPM MNI (Montreal Neurological Institute) template and then bias corrected to control for intensity differences due to inhomegeneities. The structural image was then segmented into different tissue classes (grey matter, white matter, and cerebrospinal fluid) using SPM's unified segmentation approach (*Ashburner and Friston, 2005*). The individual subjects' tissue class images from this segmentation step were used to create a custom template structural image using DARTEL (Diffeomorphic Anatomical Registration Through Exponentiated Lie Algebra). The structural images were transformed to MNI space, and finally, were smoothed with an 8-mm full-width at half-maximum (FWHM) Gaussian kernel. The functional images were motion-corrected and were realigned to the middle functional image to correct for effects of slice timing. The EPI images were then coregistered to the structural image and normalized to MNI space using the DARTEL template. Functional Images were smoothed using a 8 mm FWHM Gaussian kernel.

## fMRI general linear model

A GLM was constructed containing separate regressors corresponding to the 6 different event types: encoding of the display, vividness rating, 3 separate regressors for successfully retrieved features (color/orientation/location), as well as one regressor for unsuccessfully retrieved features (it was not possible to split the unsuccessful trials into separate features due to the low number of unsuccessful trials). To classify trials as successfully or unsuccessfully retrieved, we generated an error threshold (from 0–180 degrees) based on the model fitted across all errors, below which trials would be classed as successfully retrieved, and above which trials would be classed as unsuccessful. Specifically, the probability density functions of the von Mises and uniform distributions were computed and the threshold for successful retrieval was calculated by estimating the point at which the slope of the von Mises distribution approached zero, approximated by determining the point at which the probability that a response was drawn from the von Mises distribution reached 5% (resulting in a cut off of +/- 63 degrees from the target value). Therefore, successfully retrieved trials were defined as a response within 63 degrees of the correct answer. Even though the three features were modeled separately, all analyses were conducted across the features so that fMRI effects reflected the underlying memory processes tested, not memory for a particular feature.

In a first step, a subject-specific model was constructed in which trials were modeled by convolving a boxcar function (corresponding to the event durations and beginning at the onset of each event of interest) with the canonical hemodynamic response function. The durations used to model the data were 12 s for the encoding displays, 8 s for the vividness rating, and 6 s for the feature questions. Moreover, parametric modulators were included to model precision and vividness effects that reflect the degree to which activity correlated with trial-by-trial fluctuations in behavior. The parametric modulator for encoding trials corresponded to the number of subsequently successfully retrieved features (ranging from 0 to 6) for each display. The trial-by-trial vividness ratings (continuous from 0–100) were included as a parametric modulator for vividness trials, and, for successfully retrieved trials brain activity was modulated with the precision of the response, which was calculated as 180 minus the error so that higher values indicated higher precision (responding close to the target). Therefore, precision values ranged from 180 (maximum precision) to 117 (minimum precision

for successful retrieval). Unsuccessfully retrieved trials were not parametrically modulated as any variability in error from the target value was assumed to be due to guessing in these trials. Parametric modulators were mean centered to ensure that the non-parametric effects of these variables represent the mean activity of the event type (*Mumford et al., 2015*). Time and dispersion derivatives were included for each event and each parametric modulator. The 8 separate scan blocks were concatenated for analysis to obtain more stable parameter estimates due to increasing the number of trials per condition, as is often the case for designs splitting trials by performance (*Uncapher et al., 2006*; *Bergström et al., 2015*). As runs were concatenated, standard high pass filtering using SPM was not possible, as SPM would treat the runs as one continuous time-series. Therefore, a high pass filter was applied by including six additional regressors for each block representing a Discrete Cosine Transform (DCT) set capturing frequencies up to 1/128 Hz. Six additional regressors representing motion were included per block and eight additional constant regressors were included to model differences between the blocks. Subject-specific effects were then entered into second-level, random effects analyses.

## Regions and contrasts of interest

Our analyses focused on the posterior medial memory network (*Ranganath and Ritchey, 2012*). We chose three ROIs based on this network: the hippocampus as the core of this network, the AnG as a lateral parietal ROI, as well as the precuneus as a medial parietal ROI. Each of these regions has previously been linked with one or other of the components of memory retrieval tested in the current study. Hippocampus and AnG have been associated with objective recollection success (*Kim, 2010*), with some evidence for a dissociation in the function of these regions (*Vilberg and Rugg, 2012*). AnG was of specific interest in the current study as previous work indicates that this area might represent individual memories (*Bonnici et al., 2016*; *Kuhl and Chun, 2014*) and has been implicated in subjective aspects of retrieval (*Simons et al., 2010*; *Yazar et al., 2014*; *Kuhl and Chun, 2014*). Lastly, the precuneus has been associated with vivid memory retrieval (*St-Laurent et al., 2015*) and processes associated with mental imagery (*Fletcher et al., 1995*). Within each of our ROIs, we focused on three contrasts of interest: successful vs. unsuccessful retrieval, positive correlation with precision, using a parametric modulator of response precision for successfully retrieved trials, and positive correlation with vividness, using a parametric modulator of vividness ratings during vividness judgment trials. ROIs were generated using the Wake Forest University (WFU) pick atlas, with anatomical masks for the three ROIs selected from the automated anatomical labeling (AAL) atlas. Small-volume correction at a family-wise error (FWE) corrected threshold of $p < 0.05$ was applied when assessing the statistical significance of the ROI results.

# Acknowledgements

This study was funded by a James S McDonnell Foundation Scholar Award to JSS, and was carried out within the University of Cambridge Behavioural and Clinical Neuroscience Institute, funded by a joint award from the Medical Research Council and the Wellcome Trust. RAC was funded by the Economic and Social Research Council and PMB by the Wellcome Trust. We would like to thank the staff of the MRC Cognition and Brain Sciences Unit MRI facility for scanning assistance, Masud Husain for valuable advice, and Tim Brady for providing access to material used in the study.

# Additional information

## Funding

| Funder | Author |
| --- | --- |
| Wellcome Trust | Paul M Bays<br>Jon S Simons |
| James S. McDonnell Foundation | Jon S Simons |
| Medical Research Council | Jon S Simons |

The funders had no role in study design, data collection and interpretation, or the decision to submit the work for publication.

## Author contributions
FRR, RAC, Conception and design, Acquisition of data, Analysis and interpretation of data, Drafting or revising the article; PMB, Conception and design, Analysis and interpretation of data; JSS, Conception and design, Analysis and interpretation of data, Drafting or revising the article

## Author ORCIDs
Franziska R Richter, http://orcid.org/0000-0002-1917-0435
Rose A Cooper, http://orcid.org/0000-0003-1521-8371
Paul M Bays, http://orcid.org/0000-0003-4684-4893
Jon S Simons, http://orcid.org/0000-0002-7508-9084

## Ethics
Human subjects: Informed consent was obtained according to procedures approved by the Cambridge Psychology Research Ethics Committee.

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
