## [Decision Letter]

Thank you for submitting your article "Distinct neural mechanisms underlie the success, precision, and vividness of episodic memory" for consideration by *eLife*. Your article has been favorably evaluated by Timothy Behrens (Senior Editor) and three reviewers, one of whom, Lila Davachi (Reviewer #1), is a member of our Board of Reviewing Editors, and another one is Charan Ranganath (Reviewer #3).

The reviewers have discussed the reviews with one another and the Reviewing Editor has drafted this decision to help you prepare a revised submission.

The paper makes a much needed step forward in further clarifying the roles of distinct regions in memory retrieval and the novel methods adopted to measure continuous aspects of memory retrieval is creative and potentially impactful. However, the reviewers had several questions about particular aspects of the analyses and results which will help readers to better understand (and replicate) what has been done. Indeed, the Methods appear a bit 'thin' and should include more detailed information about the methods employed. I am appending all the reviews below since there are several questions/clarifications the reviewers requested to which you should respond in detail.

*Reviewer #1:*

This paper makes an effort to measure different aspects of recollection – success, vividness and precision by tailoring the retrieval portion of the task to assess subjects’ memory for these different aspect of the recollective experience. During each encoding phase, 6 scenes each containing 3 objects were presented for study. After this phase, a scene was presented without the objects and participants were first asked about the vividness of remembering the 4 objects associated with each scene. They were then asked to recover details about 2 of the 3 previously presented objects in a continuous manner. Using retrieval success, continuous measure of precision and vividness seemed to correlate with three distinct regions of the core recollection network: hippocampus for success, angular gyrus for precision and precuneus for vividness. The attempt is lauded and the data make a nice contribution to the literature. Questions and concerns about the analytic approach however require consideration. Also, do these components differentiate during encoding as well or are these solely seen at retrieval?

The behavioral measures were all correlated with each other but the authors make the valid argument that a substantial amount of variance is still left unexplained so the measures are capturing distinct components of the recollective process. I wonder whether retrieval success and precision should not be more correlated since they are (i think) derived from the same measure? Or am I missing something?

Success: looks like they discretize success using the model into successful and unsuccessful but then use the continuous measures of success to define 'precision'? This is an interesting analysis as the categorical way of defining the data pulls out hippocampus (consistent with prior work) while the continuous measure pulls out angular gyrus. Finally, precuneus appear sensitive to the vividness of retrieval as assessed prior to any source questions. It looks like both hippocampus and angular gyrus are sensitive to success but only angular gyrus shows a continuous effect, which is a nice new finding.

It has previously been shown that encoding activation in hippocampus tracks the number of details later remembered (Staresina and Davachi, 2008) which seems related to the current 'precision' measure. It would be interesting to see if these roles during retrieval extend to encoding or at least for some discussion of this point made in the paper.

Analyses were primarily conducted in a priori regions of the recollective network. But whole brain conjunction analyses were also performed. The threshold set was.001 uncorrected which when put into a conjunction becomes less liberal. I wonder if the authors had tried a conjoint probability of.001 (each contrast set to.01) if any more regions would have emerged. Or maybe this is what they meant they already did?

All in all, I think the paper makes a nice contribution but the details of the model are really the novel contribution here and those details are not sufficiently described in the manuscript. I also think the data seem a bit 'thin'.

Figure 3 – it is not clear to me what is shown in 3A, just the ROIs? A priori? Or the result of some sort of contrast? Also, the statistical threshold is suspect –.05 with a SVC? For simple ROI analyses, they should survive a.05 without correction, in my opinion since you are not running a contrast in each voxel in the ROIs, or are you?

*Reviewer #2:*

The Introduction is well-written and very successfully brings together features from the long-term and working-memory literature to motivate the study. Additionally, the Discussion highlights the significance of the current findings to a wide range of existing studies and does a nice job of advancing the conversation about important theoretical issues in episodic memory research.

1) My understanding of the color manipulation (and maybe the location and rotation manipulations too) is that only a limited number of distinguishable colors were used rather than the entire spectrum. Please disregard this comment if I'm wrong. If this is true though, does the rotational response for color also change in those same increments? If not, and the response is continuous, how close does it need to be to the actual color in order to be considered correct? Is this where the "+/- 63 degrees from the target value" comes in?

2) One of the main strengths of the study, from an analysis perspective, is the precise, parametric modeling of the behavioral features on the fMRI data. In my opinion though, more information about the parametric modulators would be helpful for others to implement a similar approach. In particular, for the retrieval trials, was the whole 100-point scale used for vividness, or were the ratings grouped into fewer bins? This comment also applies to my previous one about the rotational responses, where I'm a little unclear about whether those data are binned or continuous for the modulator.

3) For the analyses of behavioral data, the t-tests provide good indication of the overall pattern of how the different models fit (model 2 > 3 > 1). But if the individual subject data are looked at in more detail, do all (or most) of the subjects tend to support the same model? Or are there any subjects that seem to be doing something different? Maybe the individual subject data are too noisy to say anything definitive in this way, which is fine too.

4) The analyses used to functionally dissociate the fMRI effects seem somewhat untraditional. As I understand the current analysis, one of the follow-up ANOVAs uses the retrieval success and precision effects for the HIPP and AnG, since these regions show the largest respective effects. A significant interaction then might pick up on the retrieval success > precision effect in HIPP, but it doesn't say anything about there being a precision > retrieval success effect in AnG (the retrieval success effect is actually larger). If one wants to determine whether the AnG (or some AnG voxels) is involved more in precision than in retrieval success or vividness, wouldn't it be more appropriate to use an exclusive-masking procedure for each ROI?

*Reviewer #3:*

The present study aimed to examine the distinct contributions of a subset of regions within the recollection network when people engage in episodic retrieval. Using a paradigm that allowed the authors to dissociate three putative components (i.e., retrieval success, precision and vividness) associated with episodic retrieval, the authors reported that distinct regions of the recollection network are preferentially involved in retrieval success, retrieval precision, and subjective vividness of episodic memories. Results from this study are in line with emerging behavioral evidence, which suggests that episodic recollection may consist of separable components (e.g., retrieval success and precision). The current results thus provide additional neural evidence for these behavioral observations, and shed lights to the functional contributions of distinct brain regions within the recollection network to episodic retrieval. The current report is well written and will generate broad interest in the memory research community. We only have a few comments and questions:

1) In the current study, the lack of hippocampal involvement during subjective vividness ratings seems to differ from studies showing that hippocampal activity is associated memory vividness (e.g., Ford and Kensigner, 2016, Gilboa et al., 2004). I wonder if the authors could elaborate more on what might be contributing to the differences between studies.

2) Related to Point 1. It seems that the range of values (or variance) for vividness regressor and precision regressors are different. I wonder if this could lead to the lack of hippocampal activation in the vividness contrast.

3) Selection of ROIs. Although precuneus is part of recollection network, it seems that other regions within the network, including the retrosplenial cortex, posterioral cingulate cortex, and the medial PFC may equally warrant for investigation. I wonder if the authors could provide more information on the motivation to specifically focus on precuneus.

4) In the Discussion, the authors argued that the finding of vividness coding in the precuneus might be related to its interconnection with lateral parietal cortex/angular gyrus, whose activity has been shown to be associated with vividness ratings. (Discussion, last paragraph). This argument doesn't seem to fit with the current null finding of vividness coding in the angular gyrus. Given the relatively well-established relationship between vividness coding and activity in angular gyrus (e.g., Bonnici et al., 2016; Kuhl and Chun, 2014), more discussion on the discrepancies is warranted.

---

## [Author Response]

*The paper makes a much needed step forward in further clarifying the roles of distinct regions in memory retrieval and the novel methods adopted to measure continuous aspects of memory retrieval is creative and potentially impactful. However, the reviewers had several questions about particular aspects of the analyses and results which will help readers to better understand (and replicate) what has been done. Indeed, the Methods appear a bit 'thin' and should include more detailed information about the methods employed. I am appending all the reviews below since there are several questions/clarifications the reviewers requested to which you should respond in detail.*

*Reviewer #1:*

*This paper makes an effort to measure different aspects of recollection – success, vividness and precision by tailoring the retrieval portion of the task to assess subjects’ memory for these different aspect of the recollective experience. During each encoding phase, 6 scenes each containing 3 objects were presented for study. After this phase, a scene was presented without the objects and participants were first asked about the vividness of remembering the 4 objects associated with each scene. They were then asked to recover details about 2 of the 3 previously presented objects in a continuous manner. Using retrieval success, continuous measure of precision and vividness seemed to correlate with three distinct regions of the core recollection network: hippocampus for success, angular gyrus for precision and precuneus for vividness. The attempt is lauded and the data make a nice contribution to the literature. Questions and concerns about the analytic approach however require consideration. Also, do these components differentiate during encoding as well or are these solely seen at retrieval?*

We thank the reviewer for the suggestion of dissociating the three memory components at encoding in addition to retrieval. Please see our third response to Reviewer 1 for further details.

*The behavioral measures were all correlated with each other but the authors make the valid argument that a substantial amount of variance is still left unexplained so the measures are capturing distinct components of the recollective process. I wonder whether retrieval success and precision should not be more correlated since they are (i think) derived from the same measure? Or am I missing something?*

The behavioural measures of retrieval success and precision are derived from the model, and can vary independently. Responses are modelled as a mixture of a von Mises (circular normal) distribution centred on the target value, and a circular uniform distribution. Retrieval success refers to the proportion of trials drawn from the von Mises versus the uniform distribution, while precision refers to the concentration parameter (width) of the von Mises component. As illustrated with simulated data in Figure 7, an error distribution can be characterized by high retrieval success (first two distributions) or low retrieval success (third and fourth distribution), but with varying degrees of precision for retrieved items (high for distribution 1 and 3, and low for distribution 2 and 4). For this reason it is indeed possible for the two measures to be relatively independent and only moderately correlated, as observed in the present data. We have clarified this distinction in the subsection “Behavioral analysis”.

*Success: looks like they discretize success using the model into successful and unsuccessful but then use the continuous measures of success to define 'precision'? This is an interesting analysis as the categorical way of defining the data pulls out hippocampus (consistent with prior work) while the continuous measure pulls out angular gyrus. Finally, precuneus appear sensitive to the vividness of retrieval as assessed prior to any source questions. It looks like both hippocampus and angular gyrus are sensitive to success but only angular gyrus shows a continuous effect, which is a nice new finding.*

*It has previously been shown that encoding activation in hippocampus tracks the number of details later remembered (Staresina and Davachi, 2008) which seems related to the current 'precision' measure. It would be interesting to see if these roles during retrieval extend to encoding or at least for some discussion of this point made in the paper.*

We agree that it would be very interesting for future research to dissociate neural correlates of subsequent memory success, precision, and vividness during encoding; however, differentiating these components during encoding would require a somewhat different design to the one currently employed. This is because for each single encoding display in our design, participants are subsequently asked 6 feature questions (3 questions per object). While it is fairly straightforward to derive a measure of later retrieval success (x out of 6 items were successfully remembered), it is not clear how a measure of precision (for an entire encoding display) should be derived that is independent of retrieval success. Future studies would need to use separate encoding events that each have a single subsequent precision measure to address this question. Therefore, the present data cannot answer whether hippocampus tracks subsequent *precision* of an encoded item, but, following the reviewer’s suggestion, we were able to test whether hippocampus tracks the number of subsequently remembered (successful retrieval) features. In contrast to the finding described by the reviewer above, we did not find in our data that hippocampal activity during encoding tracked the number of features subsequently recalled. As the main focus of the current study was on retrieval processes, and the current design does not allow us to adequately test the involvement of all three processes during encoding, we have refrained from addressing the question of dissociable encoding processes in the paper.

*Analyses were primarily conducted in a priori regions of the recollective network. But whole brain conjunction analyses were also performed. The threshold set was.001 uncorrected which when put into a conjunction becomes less liberal. I wonder if the authors had tried a conjoint probability of.001 (each contrast set to.01) if any more regions would have emerged. Or maybe this is what they meant they already did?*

We thank the reviewer for highlighting that this aspect of the analysis was unclear. Yes, the significance threshold that we used was indeed a final conjoint *conjunction* threshold of p <. 001; we did not set each individual contrast to. 001 as this would indeed result in an inappropriately conservative analysis. We have clarified this in the last paragraph of the subsection “ROI activity and conjunction analyses”.

*All in all, I think the paper makes a nice contribution but the details of the model are really the novel contribution here and those details are not sufficiently described in the manuscript. I also think the data seem a bit 'thin'.*

We agree with the reviewer that it would be helpful for us to include more methodological details about the behavioural and fMRI analysis approaches. We have done so at several points in the revised paper throughout the Methods and Results sections.

*Figure 3 – it is not clear to me what is shown in 3A, just the ROIs? A priori? Or the result of some sort of contrast? Also, the statistical threshold is suspect –.05 with a SVC? For simple ROI analyses, they should survive a.05 without correction, in my opinion since you are not running a contrast in each voxel in the ROIs, or are you?*

We thank the reviewer for bringing to our attention the fact that our description of what is shown in Figure 3 was not sufficiently clear. We have edited the caption of Figure 3 (and the text referring to the figure) in the revised manuscript. Figure 3 displays the results of the three contrasts of interest (retrieval success, precision and vividness) in the ROI analysis. For visualization, brain activity was masked by ROIs that showed significant results for each contrast in Figure 3. Because the results displayed the described dissociation (retrieval success effects in Hippocampus, precision effects in AnG and vividness effects in Precuneus), the resulting figure shows activity only in these areas. Of note, an unmasked version of the results of these three contrasts (at a more liberal threshold) can be seen in Figure 4. We used small-volume correction as it is one of the standard methods of ROI analysis (Worsley et al., 1996; Poldrack, 2007). Thus, the p-value of each voxel within an ROI was corrected according to the number of voxels (tests) within each ROI (family-wise error correction). Voxels were only considered significant if their p-values were <.05, corrected.

*Reviewer #2:*

*The Introduction is well-written and very successfully brings together features from the long-term and working-memory literature to motivate the study. Additionally, the Discussion highlights the significance of the current findings to a wide range of existing studies and does a nice job of advancing the conversation about important theoretical issues in episodic memory research.*

*1) My understanding of the color manipulation (and maybe the location and rotation manipulations too) is that only a limited number of distinguishable colors were used rather than the entire spectrum. Please disregard this comment if I'm wrong. If this is true though, does the rotational response for color also change in those same increments? If not, and the response is continuous, how close does it need to be to the actual color in order to be considered correct? Is this where the "+/- 63 degrees from the target value" comes in?*

We can confirm that the colour manipulation (as well as orientation and location) did not use a limited number of distinguishable colours, but was indeed continuous. We apologize if this was ambiguous in the original manuscript and have amended the text throughout the Materials and methods section to make this aspect of our design clearer. For all three features, response space was divided into 360 increments around the circle, so that subjects could change the features in an effectively continuous manner when moving around the wheel. All three features were selected from continuous space when originally generating the displays, such that any colour/orientation/location value was possible across displays. The only restriction for the generation of the displays was that *within any one display* the colours/orientations/location of the three objects would be 62.04 degrees apart. This was the minimum distance in the location feature that would ensure that objects would not physically overlap. We applied the same restriction to the other two features for consistency. However, this restriction was only applied *within* any one display; across displays all colours, orientations and locations were possible. The “+/- 63 degrees” the reviewer mentions refers to the analysis cut-off for successful versus unsuccessful retrieval, derived from the model-based analysis of the error distribution. Thus, as the reviewer suspected, when an answer fell within 63 degrees from the target response the answer was considered to be ‘correct’ (‘successful retrieval’).

*2) One of the main strengths of the study, from an analysis perspective, is the precise, parametric modeling of the behavioral features on the fMRI data. In my opinion though, more information about the parametric modulators would be helpful for others to implement a similar approach. In particular, for the retrieval trials, was the whole 100-point scale used for vividness, or were the ratings grouped into fewer bins? This comment also applies to my previous one about the rotational responses, where I'm a little unclear about whether those data are binned or continuous for the modulator.*

We agree with the reviewer and, indeed, the other reviewers that the inclusion of more methodological details would be beneficial. The entire possible scale (0-100) was used for the parametric modulator of the vividness data, and the entire possible scale (0-180) was also used for the modulation of the precision effect. However, as the precision effect was only analysed for successfully retrieved trials (within 63 degrees of the target value), only precision values (180-error) between 117 and 180, were included. We now make these points clear in the revised manuscript (subsection “fMRI general linear model”, last paragraph).

*3) For the analyses of behavioral data, the t-tests provide good indication of the overall pattern of how the different models fit (model 2 > 3 > 1). But if the individual subject data are looked at in more detail, do all (or most) of the subjects tend to support the same model? Or are there any subjects that seem to be doing something different? Maybe the individual subject data are too noisy to say anything definitive in this way, which is fine too.*

We thank the reviewer for the interesting suggestion to consider the model fit for individual subjects. We found that all of our 20 participants followed the described pattern of model fit using BIC and 17 out of 20 followed the described pattern using AIC, and thus the effect was very consistent across subjects. AIC penalizes additional parameters less than BIC does, and for the three subjects that did not follow the same pattern, the AIC was as low or even slightly lower for model 3 (having an additional parameter, as it includes non-target errors) than model 2. However, the three subjects did not show any obviously different behavioural pattern.

*4) The analyses used to functionally dissociate the fMRI effects seem somewhat untraditional. As I understand the current analysis, one of the follow-up ANOVAs uses the retrieval success and precision effects for the HIPP and AnG, since these regions show the largest respective effects. A significant interaction then might pick up on the retrieval success > precision effect in HIPP, but it doesn't say anything about there being a precision > retrieval success effect in AnG (the retrieval success effect is actually larger). If one wants to determine whether the AnG (or some AnG voxels) is involved more in precision than in retrieval success or vividness, wouldn't it be more appropriate to use an exclusive-masking procedure for each ROI?*

The reviewer is correct in that the current analysis primarily focuses on relative/disproportionate differences in activity between ROIs for specific contrasts of interest (i.e., interactions between ROIs and brain regions) rather than dissociations within individual ROIs. In our earlier analysis where we report ROI results looking for activated voxels with small volume correction, we only found hippocampus to respond significantly to retrieval success, AnG to respond significantly to precision, and precuneus to respond significantly to vividness. Therefore, an exclusive masking procedure (e.g., of AnG activity with the retrieval success contrast) would not reveal any significant voxels as each ROI only showed significant activity for one contrast of interest. Furthermore, an exclusive masking procedure similarly would not be able to reveal a difference in the magnitude of activation between contrasts; only a difference in the location of activity. We therefore agree that we cannot make the firm conclusion that AnG responds *more* to precision than to retrieval success, and take care in the manuscript only to report that there is a dissociation between ROIs and contrasts, as shown by the interactions.

*Reviewer #3:*

*The present study aimed to examine the distinct contributions of a subset of regions within the recollection network when people engage in episodic retrieval. Using a paradigm that allowed the authors to dissociate three putative components (i.e., retrieval success, precision and vividness) associated with episodic retrieval, the authors reported that distinct regions of the recollection network are preferentially involved in retrieval success, retrieval precision, and subjective vividness of episodic memories. Results from this study are in line with emerging behavioral evidence, which suggests that episodic recollection may consist of separable components (e.g., retrieval success and precision). The current results thus provide additional neural evidence for these behavioral observations, and shed lights to the functional contributions of distinct brain regions within the recollection network to episodic retrieval. The current report is well written and will generate broad interest in the memory research community. We only have a few comments and questions:*

*1) In the current study, the lack of hippocampal involvement during subjective vividness ratings seems to differ from studies showing that hippocampal activity is associated memory vividness (e.g., Ford and Kensigner, 2016, Gilboa et al., 2004). I wonder if the authors could elaborate more on what might be contributing to the differences between studies.*

The reviewer raises an important point for discussion. We believe that one of the strengths of the current approach compared to previous studies is that those studies did not try to tease apart distinct components of memory retrieval such as vividness and retrieval success. As can be seen from the regression analyses, these components are indeed clearly dissociable. By modelling the three contrasts (retrieval success, vividness, and precision) in the same GLM, the current study ensured that they accounted for independent sources of variance, thereby making it possible to discriminate the effects of different components of the retrieval process in a manner that previous studies could not. Therefore, it is possible that previous studies reporting vividness effects in hippocampus may have been picking up on a response to retrieval success. In this sense, we see our results not as conflicting with previous results, but rather they help elucidate what might be driving vividness effects in hippocampus if they are observed. We have included this reasoning at several points in the Discussion of the revised manuscript.

*2) Related to Point 1. It seems that the range of values (or variance) for vividness regressor and precision regressors are different. I wonder if this could lead to the lack of hippocampal activation in the vividness contrast.*

The parametric vividness regressor ranged from 0 to 100, the regressor for precision ranged from 117 to 180 (as we only tested precision for successful retrieval (+/- 63 degrees) trials, and the retrieval success regressor was binary (successful vs. unsuccessful). Therefore, it could be argued that the relatively larger range of values in the vividness regressor should have increased (and not limited) our ability to detect vividness effects in hippocampus if they existed, compared to, for example, retrieval success effects. Moreover, if a difference in the range of values affected our ability to detect a correlation between neural activity and vividness ratings, we would have no reason to predict that precuneus and not hippocampus would correlate with vividness.

3) Selection of ROIs. Although precuneus is part of recollection network, it seems that other regions within the network, including the retrosplenial cortex, posterioral cingulate cortex, and the medial PFC may equally warrant for investigation. I wonder if the authors could provide more information on the motivation to specifically focus on precuneus.

We agree with the reviewer that we could conceivably have justified assessing other areas in addition to or instead of the chosen ROIs. Our ROIs were chosen *a priori* for a variety of reasons which we now highlight more clearly in the manuscript text (see subsection “Regions and Contrasts of Interest”). In short, the precuneus was chosen, firstly, due to its involvement in mental imagery (Fletcher et al., 1995), as we expected that mental imagery might be a central, dissociable component part of memory retrieval, influencing particularly the vividness with which memories are experienced. Moreover, the current study was in part inspired by the memory deficits observed in patients with parietal lesions. While lateral parietal lesions are often the focus of this neuropsychological work, many of the patients in the literature likely have damage extending to broader parietal areas including the precuneus (Berryhill et al., 2007). In order to clearly understand parietal memory effects, we therefore wanted to include this medial parietal region in addition to lateral cortex. Of note we chose a fairly broad precuneus ROI: the AAL precuneus mask used in the current study includes large parts of retrosplenial cortex as well (Rolls et al. (2015), though the reviewer is correct in that it does not include posterior cingulate cortex.

With regard to medial PFC, we agree that this area has also been of interest for memory retrieval. From the whole brain analysis, it can indeed be seen that a region in ventral mPFC appears to be active for the retrieval success contrast, at least at the very liberal threshold used in Figure 4. However, our approach focused primarily on the mentioned more posterior regions, as they have been more directly linked to the processes (retrieval success, imagery, construction and vividness) that we were trying to dissociate. While our current approach focuses on the three a priori selected ROIs, we included the exploratory whole brain analysis to obtain a better understanding of processes that might be supported by other nodes in the broader retrieval network. We believe that this combination of a small number of pre-selected ROIs together with a whole brain analysis provides an ideal combination of hypothesis driven and exploratory analyses.

*4) In the Discussion, the authors argued that the finding of vividness coding in the precuneus might be related to its interconnection with lateral parietal cortex/angular gyrus, whose activity has been shown to be associated with vividness ratings. (Discussion, last paragraph). This argument doesn't seem to fit with the current null finding of vividness coding in the angular gyrus. Given the relatively well-established relationship between vividness coding and activity in angular gyrus (e.g., Bonnici et al., 2016; Kuhl and Chun, 2014), more discussion on the discrepancies is warranted.*

The reviewer is correct that previous studies have linked AnG and subjective vividness reports, and that discussing the apparent divergence in results is of importance. The revised manuscript now includes several changes in the Discussion that address these interesting points in more detail. Our interpretation is that AnG may support the representation of retrieved information, the precision of which is utilised by the precuneus in the service of vividness judgements. Regarding the sentence in the Discussion of the original manuscript, we agree that our wording was not sufficiently clear. Our intention was to argue that the (possibly indirect) link between diminished vividness reports and (lateral) parietal lesions that has been drawn from patient studies might be caused by two factors: firstly, lesions are often not selective to lateral parietal areas, and include medial parietal areas such as precuneus as well; secondly, as medial and lateral parietal areas are heavily connected, damage to lateral parietal areas might disrupt input from these areas to precuneus, which could result in reduced vividness reports.